# Potential of *Carica papaya* Seed-Derived Bio-Coagulant to Remove Turbidity from Polluted Water Assessed through Experimental and Modeling-Based Study

Amir Hariz Amran [1], Nur Syamimi Zaidi [1,2], Achmad Syafiuddin [3], Loh Zhang Zhan [1], Muhammad Burhanuddin Bahrodin [1], Muhammad Aamer Mehmood [4] and Raj Boopathy [5,*]

[1] School of Civil Engineering, Faculty of Engineering, Universiti Teknologi Malaysia (UTM), Johor Bahru 81310, Johor, Malaysia; amir94hariz@gmail.com (A.H.A.); nursyamimi@utm.my (N.S.Z.); zhangzhanloh@gmail.com (L.Z.Z.); burhanuddinbahrodin@yahoo.com (M.B.B.)

[2] Centre for Environmental Sustainability and Water Security (IPASA), Universiti Teknologi Malaysia (UTM), Johor Bahru 81310, Johor, Malaysia

[3] Department of Public Health, Universitas Nahdlatul Ulama Surabaya, Surabaya 60237, East Java, Indonesia; achmadsyafiuddin@unusa.ac.id

[4] Department of Bioinformatics and Biotechnology, Government College University Faisalabad, Faisalabad 38000, Pakistan; draamer@gcuf.edu.pk

[5] Department of Biological Sciences, Nicholls State University, Thibodaux, LA 70310, USA

* Correspondence: ramaraj.boopathy@nicholls.edu

**Abstract:** It is important to develop renewable bio-coagulants to treat turbid water and efficient use of these bio-coagulants requires process optimization to achieve robustness. This study was conducted to optimize the coagulation process using bio-coagulant of deshelled *Carica papaya* seeds by employing response surface methodology (RSM). This bio-coagulant was extracted by a chemical-free solvent. The experiments were conducted using the Central Composite Design (CCD). Initially, the functional groups and protein content of the bio-coagulant were analyzed. The Fourier Transform Infrared Spectroscopy analysis showed that the bio-coagulant contained OH, C=O and C-O functional groups, which enabled the protein to become polyelectrolyte. The highest efficiency of the bio-coagulant was obtained at dosage of 196 mg/L, pH 4.0 and initial turbidity of 500 NTU. At the optimum conditions, the bio-coagulant achieved 88% turbidity removal with a corresponding 83% coagulation activity. These findings suggested that the deshelled *Carica papaya* seeds have potential as a promising bio-coagulant in treating the polluted water.

**Keywords:** *Carica papaya* seed; bio-coagulant; coagulation-flocculation; respond surface methodology

## 1. Introduction

The coagulation-flocculation process is a common method in wastewater treatment due to its effectiveness in removing organic matter, turbidity and color [1,2]. The conventional coagulation process involves the addition of divalent positively charged chemical compounds such as aluminum sulphate and ferric chloride, which are known to have various drawbacks in both health and environmental perspectives. In the health perspective, due to the nondegradable nature of the chemical coagulants, the chemical residues in the treated water can be accumulated in body cells for a long time [3,4] posing negative health impacts including nervous system failure, Alzheimer's disease, and dementia [5,6]. In terms of an environmental perspective, the chemical coagulants are relatively expensive and commonly cause secondary contaminations from sludge due to higher levels of residual chemicals. Ultimately, the sludge produced is toxic and considered as a scheduled waste [2,7,8].

Bio-coagulants are being developed as an alternative to the chemical coagulants and their use to treat turbid water has become imperative due to their biodegradable and environmentally friendly nature. During past few years, studies on bio-coagulants had widely

reported for their applications in wastewater treatment. Bio-coagulants have been obtained from various plant species such as *Moringa oleifera* [9,10], *Jatropha curcas* [1,11], tannin [12], banana peels [13,14] and *Bagasse* [10]. These studies have shown that bio-coagulants pose promising treatment performance and thus stand a great chance in replacing the conventional chemical coagulants. Not limited to promising treatment performance, the used bio-coagulants in water and wastewater treatment produce non-toxic and lesser volume of sludge and the bio-coagulant itself is highly degradable unlike chemical coagulants. Besides, application of bio-coagulant in water treatment can eliminate the cost of purchasing and importing chemicals as the raw materials for bio-coagulants can be obtained locally [12].

*Carica papaya*, commonly known as papaya has been a popular fruit throughout the world. It tends to grow in tropical or subtropical regions and its international trade has been recorded worth nearly 200 million US $ in 2009. Consumption of papaya fruits results in a mass production of food wastes, particularly the discarded papaya peels and seeds which constitutes 15–20% of its weight [15]. Hence, it is essential to minimize the quantity of the papaya waste by utilizing it in various aspects including as the bio-coagulant in wastewater treatment. To date, there is lack of study investigating the performance of *Carica papaya* seeds as the bio-coagulant in either water or wastewater treatment. Moreover, the chemical characteristics of the papaya seeds such as protein and functional groups which are responsible as an active coagulation agent remains indeterminate. According to Fatombi et al. [16], the fat residues that are associated with protein in the bio-coagulant increase aggregation by forming cluster thus, promotes the coagulation process. Active coagulant agent can be extracted using various solvents such as distilled water, alcohol, acid ($H_2SO_4$) and alkali (NaOH). Different solvents affect the extracted bio-coagulants differently. Among others, the used of distilled water without additional chemicals is the most preferable as the use of chemicals could alter the initial pH of the treated water or wastewater. Furthermore, chemical-free bio-coagulant produces less sludge [17].

Besides the chemical characterization, there is also lack of knowledge in terms of the interaction effects and optimization among the prevalent operating parameters of the coagulation process. Existing study on papaya seeds as bio-coagulant was mainly focused on the single effect (by applying one-factor-at-a-time approach) of either dosage or pH towards the turbidity removal [2,18]. According to Shak and Wu [19], the relationship between different parameters plays an important role in identifying the optimum conditions for the coagulation process. Response Surface Methodology (RSM) utilizes mathematical and statistical models in identifying the optimum operating parameters to achieve a higher process efficiency [18].

Therefore, this study was conducted to assess the potential of deshelled *Carica papaya* seeds as the bio-coagulant in treating the polluted turbid water. The bio-coagulant extracts obtained from deshelled *Carica papaya* seeds were characterized for protein, functional groups by Fourier Transform Infrared (FTIR) spectroscopy, and Field-Emission Scanning Electron Microscopy (FESEM). The optimization of the turbidity removal process and coagulation activity was achieved using RSM to enhance the destabilization of the pollutant particles.

## 2. Materials and Methods

### 2.1. Preparation of Synthetic Water

Synthetic kaolin water was prepared by mixing 10 g of kaolin powder with particle size ranged from 25 μm to 35 μm in 1 L of distilled water resulting in solution of pH 4 to 5. The solution was left for 24 h. After that, the sediments were removed from the solution prior to be used as the stock solution which was diluted accordingly during the jar test.

### 2.2. Preparation of Deshelled Carica Papaya Seed as Bio-Coagulant

*Carica papaya* seeds were collected from Taman Universiti's fruit markets. The seeds were prepared by initially washing and removing the surface membrane layer using a

cloth. The seeds were deshelled and then left to dry in an oven for 24 h at 50 °C. The dried seeds were crushed to a fine powder and sieved for particle uniformity. For the bio-coagulant, particles smaller than 0.4 mm were used. Primary processing was done by mixing 500 mg of deshelled *Carica papaya* seeds powder into 500 mL of distilled water. To obtain minimally processed green coagulant, deshelled *Carica papaya* seeds were used without chemical treatment. The extracted deshelled *Carica papaya* seed was centrifuged to separate the solid from liquid and further filtered based on vacuum filtration method. The filtered solution was kept in cold room to maintain its properties and labelled as stock coagulant. Stock coagulant was diluted accordingly upon usage.

### 2.3. Coagulation Assay

Chemical coagulation/flocculation experiments were carried out at room temperature in a jar test. In 2 L beakers, synthetic kaolin water was added. The synthetic kaolin water was stirred at 250 rpm rapid mixing for 3 min after the addition of deshelled *Carica papaya* seeds bio-coagulant and slight adjustment of pH. Using a 1:2 HCl (36–38%, Vetec) and NaOH (97%, Vetec) solution, the pH was balanced to the desired value. After that, the speed of the stirring was reduced to 30 rpm for 15 min, followed by 30 min of sedimentation. Turbidity samples were taken from 5 cm below the surface after the sedimentation period and were measured immediately.

### 2.4. Characterization of Deshelled Carica Papaya Seeds

The soluble protein content in the bio-coagulant was measured based on Lowry method [20] using bovine serum albumin (BSA) as the standard. Standard curve of BSA was obtained by preparing different BSA standard solutions (20, 40, 60, 80 and 100 mg/L) from BSA stock solution of 100 mg/L. The stock solution was prepared by mixing 0.05 g of BSA into 500 mL of dilution water. Lowry solution that comprised a mixture of Solution X, Solution Y and Solution Z (Table 1) was then prepared.

**Table 1.** Composition of Solution X, Solution Y and Solution Z.

| Solution | Composition |
|----------|-------------|
| Solution X | 2.9 g of sodium hydroxide + 14.3 g of sodium carbonate |
| Solution Y | 1.4 g of copper sulfate + 100 mL of distilled water |
| Solution Z | 2.9 g of sodium tartrate + 100 mL of distilled water |

A volume of 0.5 mL of each standard solution was transferred into 10 mL glass test tubes separately, then 0.7 mL of Lowry solution was added to each tube. These mixtures were vortexed and allowed to incubate for 20 min at room temperature. During the last 5 min, Folin reagent was prepared by mixing 5 mL of 2N Folin and Ciocalteu's Phenol reagent into 6 mL of dilution water. After 20 min of incubation, 0.1 mL of Folin reagent was added to each tube. Then, the mixture was vortexed and incubated for 30 min. Finally, the absorption for all tubes and the blank (distilled water with the reagents) were measured at $\lambda 750$ nm using DR6000 Spectrophotometer (HACH). A standard curve was plotted with concentrations versus absorption. From the linear equation, the total protein concentration was calculated as BSA-equivalent compounds. In order to determine the protein content of the bio-coagulant, the above steps were repeated by replacing the standard solution with the same volume of bio-coagulant (0.5 mL). Protein concentration was determined from the calibration curve obtained with BSA as a standard. The results of each analysis were averaged after they were completed in triplicate.

The absorption spectrum of deshelled *Carica papaya* seed was obtained using FTIR (Shimadzu FTIR 8400, Kyoto, Japan). The sample was prepared by mixing the extract of deshelled *Carica papaya* seed with KBr in a 1:200 ($w/w$) ratio. A hydraulic press was used to compress the dried mixture, resulting in a homogeneous sample/KBr disc. The spectrum was recorded in the region of 500 to 4000 cm$^{-1}$. FESEM was used to analyze the surface morphology and elemental composition of the deshelled *Carica papaya* seed, which is useful

in explaining the mechanisms of coagulation. A scanning electron microscope (Hitachi TM3000, Quorum Sputter Coater, Tokyo, Japan) with an image capture system reproduced images at $500\times$ magnification.

### 2.5. Analysis of Turbidity and Coagulation Activity

A portable turbidimeter (Milwaukee turbidimeter) was used to determine turbidity. Coagulation activity was determined via jar test. A total of 500 mL of synthetic turbid water with various turbidities and pH was poured in a beaker and mixed for 3 min at 250 rpm followed by 15 min of 30 rpm. Various coagulant dosages ranging from 26–223 mg/L were added for each beaker. The suspension was then left for 30 min. After that, samples were collected from the top of the beaker (5 cm from the water surface) and turbidity was measured. Control coagulation (blank) was prepared at the same conditions but without the addition of coagulant. Coagulation activity was calculated using Equation (1).

$$\text{Coagulation activity }(\%) = \frac{\text{Final Turbidity Blank} - \text{Final Turbidity Sample}}{\text{Final Turbidity Blank}} \times 100 \quad (1)$$

### 2.6. Response Surface Methodology Based Experimental Design

Response surface methodology (RSM) via Central Composite Design (CCD) in the Design Expert software was used to optimize the coagulant dosage, pH of the medium, and initial turbidity (independent variables) for removal of turbidity and coagulation activity (dependent variables) by means of coagulation-flocculation using the bio-coagulant deshelled *Carica papaya* seeds. The dependent variables were predicted based on the Equation (2).

The CCD consisted of a duplication of $2^3$ factorial design, complemented by six axial points and five central points; a total of 33 tests were performed randomly. After reviewing the literature and conducting the preliminary trials, the high and low levels were then selected. The coagulant dosage levels, pH, and initial turbidity were studied (Table 2). For pH, initially, the factorial design was run to obtain the Pareto effect. In this run, pH 3.0 and 7.0 were used. For CCD analysis, minor revision was done on pH level. The pH was slightly changed to pH 4.0 and 6.5, taking into consideration for more feasible axial point $(-\alpha)$. If the low range of pH 3.0 was continuously used in the subsequent CCD analysis, the $-\alpha$ can be about pH 2.0, which then can potentially deteriorate the bio-coagulant.

$$Y = \beta_o + \sum_{i=1}^{n} \beta_i X_i + \sum_{i=1}^{n} \beta_{ii} x^2{}_i + \sum_{i=1}^{n} \sum_{j=i+1}^{n} \beta_{ij} x_i x_j + \varepsilon \quad (2)$$

where,

Y = Predicted responses (turbidity removal and coagulation activity)

$\beta_o$ = Intercept coefficient;

$\beta_i$ = Linear coefficient;

$\beta_{ii}$ = Quadratic coefficient;

$\beta_{ij}$ = Interactive coefficient; $x_i x_j$ = Influenced factors;

$\varepsilon$ = Random error.

The effect of the variables coagulant dosage, pH and initial turbidity were evaluated by analysis of variance (ANOVA). The order of statistical significance along with the model's hierarchy criterion were used as a basis to determine either the terms would or would not remain in the prediction models. After attaining the prediction models, three-dimensional (3D) response surface plots were generated to visualize and study the effects of the variables and the interactions.

**Table 2.** Coded and real levels of independent variables in the CCD.

| Independent Variables | Level of Independent Variables | | | | |
|---|---|---|---|---|---|
| | $-\alpha$ | $-1$ | 0 | +1 | $+\alpha$ |
| A: Dosage (mg/L) | 26.3 | 50 | 125 | 200 | 223.7 |
| B: pH | 3.6 | 4.0 (3.0) * | 5.3 | 6.5 (7.0) * | 6.9 |
| C: Initial Turbidity (NTU) | 36.8 | 100 | 300 | 500 | 563.2 |

* pH used in the factorial analyses to obtain the Pareto effect.

## 3. Results and Discussion

### 3.1. Characterization of Bio-Coagulant of Deshelled Carica Papaya Seeds

3.1.1. Protein Content

The active coagulant agents from grounded solid samples can be extracted from deshelled seeds by water, or even other extractants such as salt solutions, buffer solutions or organic solvents [17,21]. In this paper, the extraction efficiency of distilled water was studied to determine the influence of water as a solvent. Obtained extracts were analyzed and protein content (mg/mL) was determined.

Protein in the extract of deshelled *Carica papaya* seeds was present in comparable concentration with other established bio-coagulants (Table 3). The recorded protein content of deshelled *Carica papaya* seeds is 0.363 mg/mL, slightly lower than the protein in the extract of *Quercus robur* (0.540 mg/mL). The protein content of deshelled *Carica papaya* seeds was also comparable to the study by Amaglo et al. [22] that obtained protein extract from an established bio-coagulant of *Moringa oleifera* with an average of 0.371 mg/mL. The only difference is the protein from *Moringa oleifera* was extracted using organic solvent while for the deshelled *Carica papaya* seeds, the protein was extracted using distilled water, without influence of chemical addition. This makes the extraction method employed in this study, eco-friendlier and more cost-effective. These findings suggested that the deshelled *Carica papaya* seeds could possibly have higher concentration of natural polyelectrolyte (which in this case is protein) if further extraction is conducted by using chemical extractants. Nonetheless, the deshelled *Carica papaya* seeds still pose sufficient coagulation ability as other established bio-coagulant viz. *Moringa oleifera*.

**Table 3.** Comparison of protein content comprised bio-coagulant deshelled *Carica papaya* seeds and other bio-coagulants.

| Bio-Coagulant | Protein (mg/mL) | With or W/out Purification | References |
|---|---|---|---|
| Deshelled *Carica papaya* seeds | 0.363 | No | This study |
| *Quercus robur* | 0.540 | No | Antov et al. [23] |
| *Phaseolus vulgaris* | 0.081 | Yes | Petrovic et al. [24] |
| *Moringa oleifera* | 0.371 | Yes | Amaglo et al. [22] |
| *Moringa oleifera* | 0.739 | Yes | Okuda et al. [25] |
| Chestnut | 0.420 | Yes | Šćiban et al. [26] |
| Acorn | 0.440 | Yes | |

3.1.2. FTIR Spectroscopy

Effectiveness of natural polyelectrolytes in destabilizing the colloidal particles rely on the degree of ionization and copolymerization of the functional groups. Additionally, its effectiveness also depends on the quantity of substituted groups present in that structural polymer [27,28]. The functional group in the deshelled *Carica papaya* seeds powder bio-coagulant were identified in the FT-IR spectrum range 500–4000 cm$^{-1}$ (Figure 1). The 3000 cm$^{-1}$ peak was linked to -OH of polysaccharides groups [29] while the band at 2925 cm$^{-1}$ and 2855 cm$^{-1}$ are relative to the stretching of OH groups band to methyl group (C-OH) [30]. Wavenumber 1747 cm$^{-1}$ corresponded to C=O carbonyl group of carboxylic acid whereas 1657 cm$^{-1}$ which indicates stretching of carboxylic acid (-C=O) with inter-molecular hydrogen bond [29]. The bands at 1466 cm$^{-1}$ and 1379 cm$^{-1}$ were

attributed to -C-O group [30]. Bands at 1242 cm$^{-1}$ and 1163 cm$^{-1}$ were relative to ether, ester or phenol groups [31].

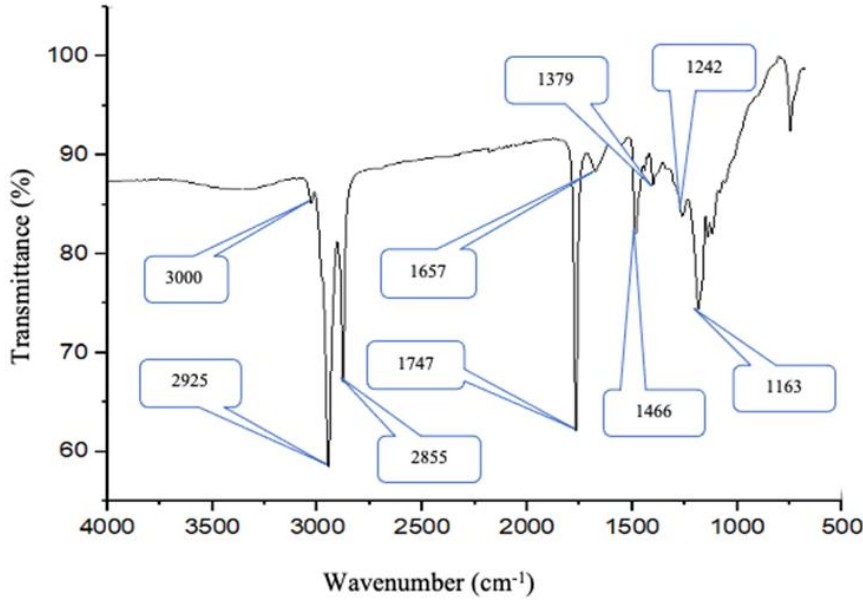

**Figure 1.** FTIR spectra of bio-coagulant deshelled *Carica* papaya seeds.

The presence of a large number of functional groups in deshelled *Carica papaya* seeds powder indicates its potential to adsorb a wide range of contaminants and, therefore, capable of facilitating the removal of suspended and dissolved substances from water. The FTIR spectra analysis revealed that OH, C=O and C-O are the dominant functional groups of the studied bio-coagulant which enabled the protein to be an efficient polyelectrolyte. In addition, the charge density of deshelled *Carica papaya* seed was recorded as a weak-positive charge of +0.4 meq/g (polycation), which strengthen the possibility of the deshelled *Carica papaya* seeds in becoming potential bio-coagulant. The negatively charged colloids may interact with positively charged bio-coagulant for active coagulation activity via charge neutralization as the coagulation mechanism.

### 3.1.3. Surface Morphology

Surface morphology of the deshelled *Carica papaya* seeds was observed by FESEM. Figure 2 shows the image obtained at 500X magnification. It is observed that deshelled *Carica papaya* seeds have heterogeneous morphology. The presence of pores on the surface of deshelled *Carica papaya* seeds can be observed, which indicate possible sites for the adsorption of particles. This surface structure can facilitate ion absorption due to its intervening small spaces with the presence of protein compound [32]. The porous structure also eases the oil extraction from the *Carica papaya* seeds [33] as well as enabling the seed to perform adsorption for charge neutralization [34].

The rough surface with the porous space, according to Subramonian et al. [35] indicates the existence of a larger surface region, which may help to provide more adsorption sites during the coagulation-flocculation process. The large number of adsorption sites on the surface of deshelled *Carica papaya* seeds can also improve the mechanism of charge neutralization. As a consequence, increased coagulation efficiency could be possible [36,37].

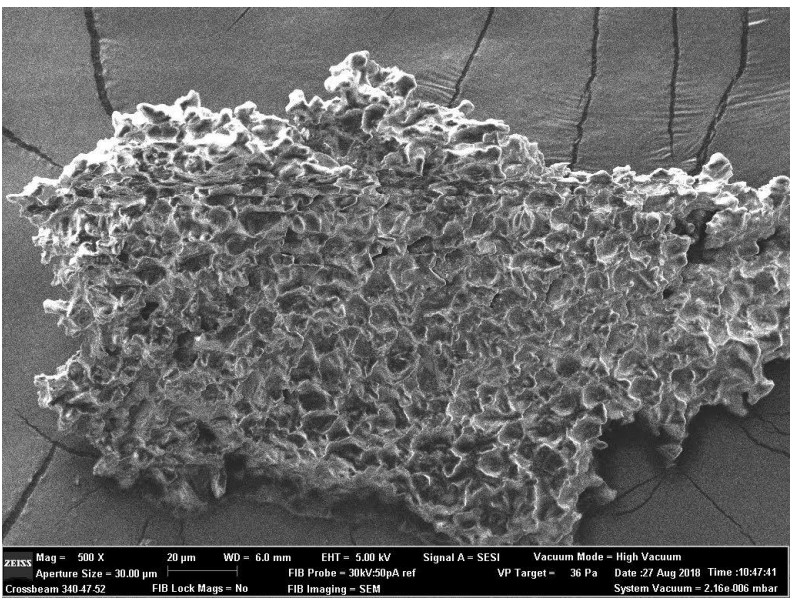

**Figure 2.** FESEM microphotograph of deshelled *Carica papaya* seeds at 500× magnification.

### 3.2. Statistical Analysis

3.2.1. Pareto Effects

Table 4 shows the turbidity removal efficiency and coagulation ability observed in each run. The best result for the removal of turbidity, and coagulation ability from the synthetic kaolin water was obtained in run 6, where a dosage of 50 mg/L of deshelled *Carica papaya* seeds at pH 3 was used for initial turbidity of 500 NTU. In this condition, the removal efficiency of turbidity, and coagulation activity were 97.2 and 73.1%, respectively.

**Table 4.** Turbidity removal efficiency and coagulation activity from synthetic kaolin water using deshelled *Carica papaya* seeds.

| Run | Dosage (mg/L) | pH | Initial Turbidity (NTU) | Turbidity Removal (%) | Coagulation Activity (%) |
|---|---|---|---|---|---|
| 1 | 50 | 3 | 500 | 97.2 | 72.8 |
| 2 | 200 | 7 | 100 | 2.9 | 1.0 |
| 3 | 50 | 3 | 100 | 3.7 | 2.4 |
| 4 | 50 | 3 | 100 | 0.0 | 2.3 |
| 5 | 200 | 7 | 500 | 10.6 | 14.5 |
| 6 | 50 | 3 | 500 | 97.2 | 73.1 |
| 7 | 200 | 3 | 100 | 2.1 | 2.2 |
| 8 | 200 | 7 | 500 | 15.1 | 17.3 |
| 9 | 50 | 7 | 100 | 2.7 | 6.1 |
| 10 | 200 | 7 | 100 | 2.0 | 6.0 |
| 11 | 50 | 7 | 100 | 6.1 | 4.9 |
| 12 | 200 | 3 | 500 | 75.5 | 30.5 |
| 13 | 50 | 7 | 500 | 4.8 | 8.6 |
| 14 | 200 | 3 | 100 | 11.8 | 7.6 |
| 15 | 200 | 3 | 500 | 72.7 | 57.1 |
| 16 | 50 | 7 | 500 | 1.2 | 1.4 |

The Pareto charts (Figure 3) indicate the variables that are significant in the removal process of turbidity, and coagulation activity from the synthetic kaolin water. Based in the charts (Figure 3), the statistical importance of each factor on the turbidity removal, and coagulation activity from the synthetic kaolin can be compared. The interaction between all three factors viz. bio-coagulant dosage, pH and initial turbidity is the most important factor that influenced the turbidity removal and coagulation activity of the synthetic kaolin

water. As for coagulation activity alone, pH is the most significant influence factor. The increment of pH decreases the coagulation activity of deshelled *Carica papaya* seeds towards synthetic kaolin water.

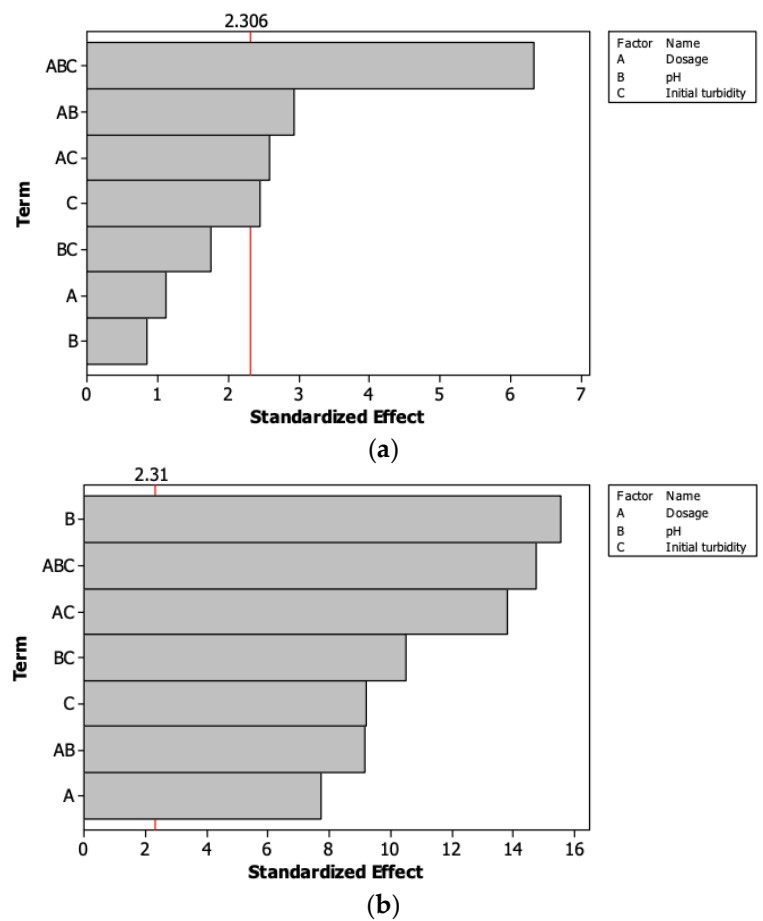

**Figure 3.** Pareto effects that correspond to regression performed for (**a**) turbidity removal and (**b**) coagulation activity from the synthetic kaolin water, *p*-value = 0.05.

The effect of increasing the deshelled *Carica papaya* seeds dosage to the treatment process was shown to be negative for turbidity removal and coagulation activity. In case of turbidity removal, the individual effect from dosage was insignificant unless it is interacted with other operational factors. Meanwhile, for the coagulation activity, even though the dosage was found significant, its effect had small relevance when compared to the pH and initial turbidity effect (Figure 3b). This suggests that a low dosage of deshelled *Carica papaya* seeds is required for the treatment of turbid water. Excessive bio-coagulant dosage can reverse the surface charge of colloidal particles, which causes dispersion and subsequently decreases the coagulant efficiency if it is not optimized well with other interacting factors [38,39].

### 3.2.2. Regression Models: Generation, Calibration and Analysis of Variance

Equations (3) and (4) show the generated second-order polynomial models with respect to the turbidity removal, and coagulation activity as a function of deshelled *Carica papaya* seeds dosage, pH and initial turbidity, respectively. Table 5 shows the analysis of variance (ANOVA), as well as the adjustment indicator coefficients. Based on the table, ANOVA for the turbidity removal shows that most terms are significant ($p \leq 0.05$) excluding square term of bio-coagulant dosage and initial turbidity, as well as interaction square term of bio-coagulant dosage and pH. Meanwhile, for the coagulation activity, ANOVA results also show that almost all terms are significant ($p \leq 0.05$) except for square

term of bio-coagulant dosage and pH. The lack of fit (LOF) for turbidity removal and coagulation activity are 0.2319 and 0.0648, respectively. The LOF for all models obtained is not significant ($p > 0.05$) thus, indicating that the models fit well with the data.

$$\text{Turbidity removal (\%)} = 876.47738 - 4.4455 \cdot A - 292.89066 \cdot B - 0.20255 \cdot C + 1.4685 \cdot AB + (4.1124 \cdot 10^{-3}) \cdot AC$$
$$+ 0.03401 \cdot BC + 23.9838 \cdot B^2 - (4.38267 \cdot 10^{-4}) \cdot ABC - (4.52659 \cdot 10^{-6}) \cdot A^2 C - 0.11940 \cdot AB^2 \tag{3}$$

$$\text{Coagulation activity (\%)} = 227.95931 - 1.44168 \cdot A - 26.09152 \cdot B - 0.49899 \cdot C + 0.022251 \cdot AB + (6.97200 \cdot 10^{-3}) \cdot AC$$
$$+ 0.034482 \cdot BC + (2.62032 \cdot 10^{-4}) \cdot C^2 - (4.40721 \cdot 10^{-4}) \cdot ABC - (1.01590 \cdot 10^{-3}) \cdot A^2 B - (1.58877 \cdot 10^{-5}) \cdot A^2 C - (5.79928 \cdot 10^{-3}) \cdot AB^2 \tag{4}$$

where,

A = Deshelled *Carica papaya* seeds bio-coagulant dosage
B = pH
C = Initial turbidity of the Synthetic Kaolin Water

The determination coefficients ($R^2$) values were shown to be 96.9% and 95.3% for turbidity removal and coagulation activity, respectively. The values of the adjusted determination coefficients (adjusted $R^2$) were 94.7%, and 92.1% for turbidity removal, and coagulation activity, respectively. An adjusted $R^2$ indicates the percentage of variation in response to an adjusted model for the number of predictors relative to the number of observations. According to Sabeti et al. [40], when evaluating the model adequacy, the discrepancy between $R^2$ and adjusted $R^2$ should be less than 20%.

**Table 5.** Developed adjusted model ANOVA for turbidity removal and coagulation activity.

| Term | Turbidity Removal | Coagulation Activity |
|------|------|------|
| | *p*-Value | *p*-Value |
| A: Dosage | 0.0005 | 0.0014 |
| B: pH | <0.0001 | 0.0114 |
| C: Initial turbidity | <0.0001 | <0.0001 |
| A × B | 0.0015 | 0.0035 |
| A × C | 0.0001 | 0.0003 |
| B × C | 0.0233 | 0.0400 |
| A × A | 0.5013 | 0.0575 |
| B × B | <0.0001 | 0.4608 |
| C × C | 0.2898 | 0.0010 |
| A × B × C | 0.0010 | 0.0022 |
| A × A × B | 0.1196 | 0.0009 |
| A × A × C | <0.0001 | <0.0001 |
| A × B × B | 0.0017 | 0.0041 |
| Lack of fit (LOF) | 0.2319 | 0.0648 |
| $R^2$ value | 96.9% | 95.3% |
| Adjusted $R^2$ value | 94.7% | 92.1% |

*3.3. Response Surface: Effect of Deshelled Carica Papaya Seeds Bio-Coagulant Dosage, pH and Initial Turbidity*

Mathematical models (Equations (3) and (4)) have generated response surfaces (Figures 4 and 5), which show the effect of deshelled *Carica papaya* seeds bio-coagulant dosage, pH and initial turbidity for turbidity removal and coagulation activity. Figure 4a shows the response surface between significant interaction of bio-coagulant dosage and pH towards the turbidity removal at fixed initial turbidity of 300 NTU. Based on Figure 4a, as the dosage increased from 50 mg/L to 200 mg/L, the turbidity removal was slightly increased. Contrary, as the pH decreased from 6.5 to 4.0, the turbidity removal was greatly increased, more significantly compared to the effect by the increased dosage. By examining the impact of pH, it is confirmed that the adjustment was crucial in order to achieve high turbidity removal performance. Since no floc formation was observed in the alkaline medium, the turbidity in the synthetic kaolin water remained nearly constant. The active compounds of the deshelled *Carica papaya* seeds coagulant extract filled with surface

charges explains the strong effect of pH on coagulation activity, which influences the interaction of bio-coagulant with the suspended particles [41,42]. In combination of the dosage and pH variables, the turbidity removal was increased to the maximum value of 71%.

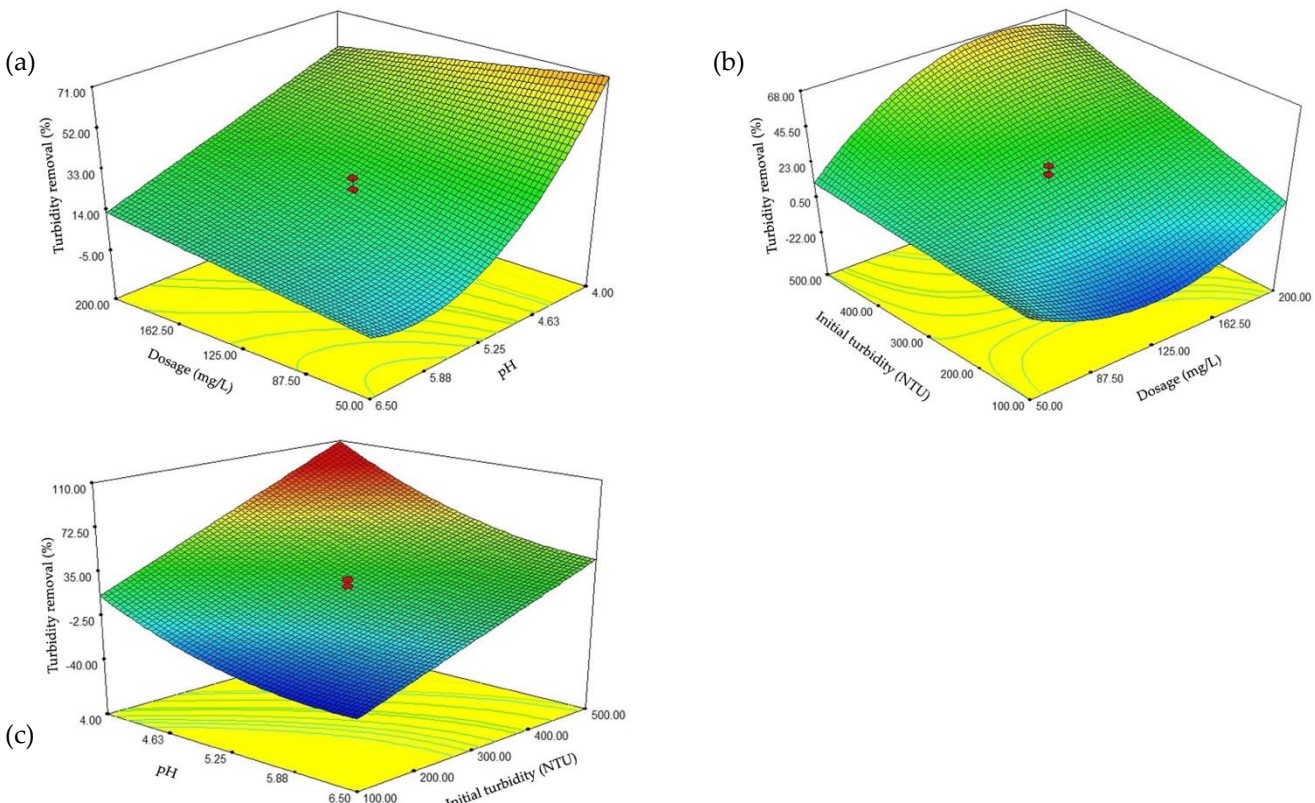

**Figure 4.** Response surface of the turbidity removal as a function of (**a**) deshelled *Carica papaya* seeds dosage and pH, (**b**) deshelled *Carica papaya* seeds dosage and initial turbidity and (**c**) pH and initial turbidity.

The response surface of turbidity removal as a function of deshelled *Carica papaya* dosage and initial turbidity is shown in Figure 4b. The surface plot illustrated the effect of wide range of bio-coagulant dosage and initial turbidity at fixed pH of 5.3. It can be seen at low dosage of 50 mg/L, an increase in initial turbidity showed a gradual increment of turbidity removal from about 2.7% to 12.0%. Contrarily, as shown in Figure 4b, the turbidity removal was positive in response to increasing dosage of the deshelled *Carica papaya* seeds when treating 500 NTU initial turbidity of the synthetic water. However, the highest removal was not observed at the maximum dosage studied, which means that by increasing the dosage is viable up to approximately 160 mg/L. An excess of coagulant above this value causes colloidal particles to stabilize, raising the residual turbidity thus, reducing the turbidity removal. In conjunction with these two variables viz. dosage and initial turbidity, the turbidity removal was able to reach up to 67% at 160 mg/L dosage and the initial turbidity of 500 NTU.

Contradictory observation on turbidity removal was shown between the interaction of initial turbidity and pH (Figure 4c). At low pH of 4, the turbidity removal was increased from 14.9% to nearly 100.0% when the initial turbidity was increased from 100 NTU to 500 NTU with the coagulant dosage kept constant at 125 mg/L. The similar positive trend was observed at high pH of 6.5 where the turbidity removal was increased up to 42.9%. Contrarily, at low initial turbidity of 100 NTU, the turbidity removal experienced a significant decrease when the pH was increased from 4.0 to 6.5. The same trend was observed at high initial turbidity of 500 NTU. The turbidity removal decreased from about 90.0% to 43.0% upon the increment of pH.

Figure 5a shows the response surface plots on the significant relationship between the bio-coagulant dosage and pH towards the coagulation activity at a constant initial turbidity of 300 NTU. Based on the figure, at low dosage of 50 mg/L, the coagulation activity was greatly increased as the pH decreased from 6.5 to 4.0. However, at high dosage of 200 mg/L, the change of coagulation activity showed a concave shape profile as the pH decreased from 6.5 to 4.0. This profile suggested that the coagulation activity was increased until a certain point, then it starts to decrease. Similar observation was seen when the dosage was increased. At pH 6.5, the coagulation activity was increased as the dosage increased from 50 mg/L to 200 mg/L. The coagulation activity increased gradually until the dosage reached approximately 140 mg/L. Beyond this dosage, the coagulation activity started to decrease. Restabilization of colloidal particles may take place at this moment, which in turn resulted in low efficiency of coagulation activity. In contrast, the coagulation activity showed a convex shape profile at low pH (pH 4) when the bio-coagulant dosage was increased. The coagulation activity was observed to be decreased until a certain point before it started to increase gradually.

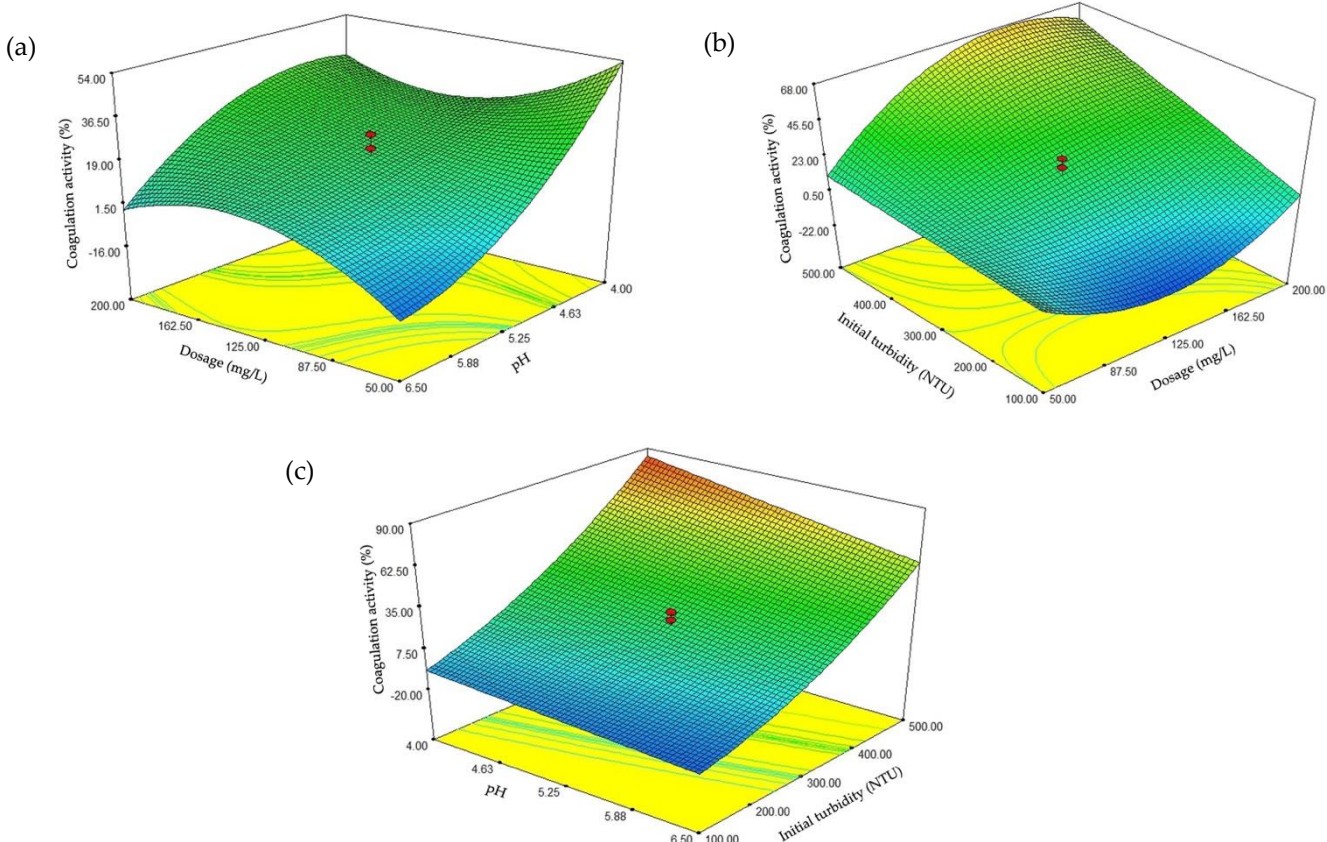

**Figure 5.** Response surface of the coagulation activity as a function of (**a**) deshelled *Carica papaya* seeds dosage and pH, (**b**) deshelled *Carica papaya* seeds dosage and initial turbidity and (**c**) pH and initial turbidity.

Another significant observation on coagulation activity was observed between the interaction of deshelled *Carica papaya* dosage and initial turbidity, at a constant pH of 5.3 (Figure 5b). Based on Figure 5b, different observation was shown upon the increase of initial turbidity from 100 NTU to 500 NTU for both low and high dosage. At low dosage of 50 mg/L, the coagulation activity was slowly decreased until the initial turbidity reached up to 250 NTU. Beyond this concentration, the coagulation activity started to show positive results. Low initial turbidity was corresponded with less colloidal particles in the water to be treated. This reduces the chance of agglomeration especially when low dosage was used. Contrary observation was seen when high dosage of 200 mg/L was used. The

coagulation activity showed consistent gradual increment as the initial turbidity increased from 100 NTU to 500 NTU.

Figure 5b also shows the increased efficiency of coagulation activity as the coagulant dosage increased up to 150 mg/L at high initial turbidity of 500 NTU. With further increment beyond this threshold value, negative efficiency was obtained. Excessive dosing has only increased the residual turbidity as polymer chain of deshelled *Carica papaya* seeds were mostly linking with one another due to surface saturation and the overloaded effect [43]. Thus, a reverse effect was observed. Equally, reductions in the bare surface area for attachment of segments following the excess of that bio-coagulant addition would lead to particle restabilization and to a certain extent, steric repulsions may form [44].

Figure 5c shows the response surface plot that illustrates the significant relationship between pH and initial turbidity while the coagulant dosage is constant at 125 mg/L. As the initial turbidity increased from 100 NTU to 500 NTU, the coagulation activity showed gradual increase at various pHs. Besides, this figure also shows that as the pH was decreased from pH 6.5 to pH 4 the turbidity increased, resulting in the slight increase in the coagulation activity. Combination between optimum pH and initial turbidity at pH 4 and 500 NTU, respectively, resulted in 80% coagulation activity.

### 3.4. Optimization of Experimental Conditions and Validation of the Models

The optimization of the deshelled *Carica papaya* seeds dosage, pH and initial turbidity for turbidity removal and coagulation activity was performed using the desirability function. The best combination found was found using deshelled *Carica papaya* seeds dosage of 196 mg/L at pH 4 with initial turbidity of synthetic kaolin water of 500 NTU. Under this condition, the turbidity removal and coagulation activity predicted by the adjusted regression models were 93.0% and 85.0%, respectively, with a desirability of 0.9596. The results from the model were compared to those obtained through experimentation, and the relative error was determined. A result of 88.0% turbidity removal and 83.0% coagulation activity were recorded against the predicted optimum responses. It can be evidenced that the mathematical models are able to satisfactorily predict the turbidity removal, and coagulation activity with the relative error between the predicted values and the values obtained in the experiment was $\pm 5\%$. This observation validates and confirms the robustness of the models that have been generated.

In comparison to the tannin extract bio-coagulant, the required optimum dosage was low; 50 mg/L for the diethanolamine (DEA) modification and 25 mg/L for the ethanolamine (ETH) modification, resulted to the >90% turbidity removal [12]. Higher dosage required by *Carica papaya* seed is probably due to the less extracted active coagulant agent. As mentioned earlier, this bio-coagulant was extracted chemical-free, only using the distilled water. Ability of the distilled water to extract active coagulant agent from deshelled *Carica papaya* seeds may not be as good as other solvents or chemical modifications but sufficient in resulting in high turbidity removal and coagulation activity exceeded 90%. In general, this study contributes significantly on the encouragement of the use of natural resources for the removal of polluted waters [45–52].

### 4. Conclusions

The bio-coagulant extracted from deshelled seeds of *Carica papaya* has shown to be a novel promising bio-coagulant to treat the synthetic kaolin water. Deshelled *Carica papaya* seeds extract was used without chemical modification/purification. The protein content of deshelled *Carica papaya* seeds was 0.363 mg/mL, which is comparable to many other established bio-coagulants. The dominant functional groups of deshelled *Carica papaya* seeds were OH, C=O and C-O, which enabled it to become polyelectrolyte. These functional groups aid in the coagulation mechanisms of deshelled *Carica papaya* seeds. The optimal condition for deshelled *Carica papaya* seeds bio-coagulant was obtained at coagulant dosage of 196 mg/L, pH 4 and initial turbidity of 500 NTU. The removal efficiency of turbidity and coagulation activity observed at this condition was 88.0% and 83%, respectively. The

use of deshelled *Carica papaya* seeds as bio-coagulant in the water treatment has proven to be an efficient, eco-friendly and feasible to replace the chemical coagulants being used in the wastewater treatment.

**Author Contributions:** Conceptualization, A.H.A. and N.S.Z.; methodology, A.H.A. and N.S.Z.; software, A.H.A.; validation, A.H.A. and N.S.Z.; formal analysis, A.H.A.; investigation, A.H.A.; resources, A.H.A.; data curation, A.H.A.; writing—original draft preparation, A.H.A. and N.S.Z.; writing—review and editing, N.S.Z., A.S., L.Z.Z., M.B.B., M.A.M. and R.B.; visualization, A.H.A. and N.S.Z.; supervision, N.S.Z., A.S. and R.B.; project administration, A.H.A. and N.S.Z.; funding acquisition, N.S.Z. All authors have read and agreed to the published version of the manuscript.

**Funding:** This study was funded by the Ministry of Higher Education (MOHE) Malaysia, grant number FRGS/1/2019/TK01/UTM/02/11) and Universiti Teknologi Malaysia (UTM), grant number Q.J130000.2651.16J76.

**Institutional Review Board Statement:** Not applicable.

**Informed Consent Statement:** Not applicable.

**Data Availability Statement:** Not applicable.

**Acknowledgments:** The authors also thank the Universitas Nahdlatul Ulama Surabaya for facilitating the research works.

**Conflicts of Interest:** The authors declare no conflict of interest.

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
