# Peer review of "Potential of Carica papaya Seed-Derived Bio-Coagulant to Remove Turbidity from Polluted Water Assessed through Experimental and Modeling-Based Study"

_applsci, doi:10.3390/app11125715_

Round 1

Reviewer 1 Report

In this study, turbidity removal was optimized using experimental design and novel biocoagulant. Manuscript has been well and clearly written but it lacks some important information and comparison to literature. Specific comments are:

- Lines 47-53: There is not much information about the biocoagulants. Authors could add some benefits of using biocoagulants, such as wide pH range of some biocoagulants. Also tannin coagulants could be mentioned, for consideration: Separation and Purification Technology, Volume 242, 1 July 2020, 116765, https://doi.org/10.1016/j.seppur.2020.116765

- The benefit of biocoagulant of this study is that there is no need for chemical treatment; this could be highlighted more clearly in the abstract and introduction.

- Section 2.1: add more information for kaolin (particle size range, quality, pH and turbidity of the supernatant).

- It is suggested run a settling experiment and follow turbidity from certain height in order to evaluate the settling performance of flocs (during the 30 min of period).

- Comparison of biocoagulant dosages needed for kaolin suspension is suggested, for example comparing the efficiency to tannin coagulant.

- What is the charge density of the biocoagulant?

- Fig. 2 has only one graph, revise the caption.

- Does this biocoagulant leach some compounds to the treated water? Have you analysed TOC of treated water?

- To discuss more deeply about mechanisms and possible restabilization, zeta potential or total surface charge analysis would be useful, if these are available for authors.

Author Response

RESPONSE TO REVIEWERS

Reviewer #1

In this study, turbidity removal was optimized using experimental design and novel bio-coagulant. Manuscript has been well and clearly written but it lacks some important information and comparison to literature. Specific comments are:

Our response: The authors thank the reviewer for the positive comment and all comments have been addressed point by point

Lines 47-53: There is not much information about the bio-coagulants. Authors could add some benefits of using bio-coagulants, such as wide pH range of some bio-coagulants. Also tannin coagulants could be mentioned, for consideration: Separation and Purification Technology, Volume 242, 1 July 2020, https://doi.org/10.1016/j.seppur.2020.116765

Our response: The authors thank the reviewer for the positive comments and suggestion. The authors have added the benefits of bio-coagulants over the chemical coagulants by referring to the suggested publication in lines 53-57.

The benefit of bio-coagulant of this study is that there is no need for chemical treatment; this could be highlighted more clearly in the abstract and introduction.

Our response: The authors have highlighted more clearly on the chemical-free of the bio-coagulant in the Abstract and Introduction in lines 22-23 and 69-74, respectively.

Section 2.1: Add more information for kaolin (particle size range, quality, pH and turbidity of the supernatant).

Our response: The authors have added more information on the kaolin in term of particle size range and pH. The turbidity of the supernatant was not studied and only initial turbidity was determined by diluting the stock solution. The additional sentences can be found in lines 93-96.

It is suggested run a settling experiment and follow turbidity from certain height in order to evaluate the settling performance of flocs (during the 30 min of period).

Our response: The authors thank the reviewer for the comment and suggestion. The authors acknowledged the importance of settling performance using Carica papaya seeds as bio-coagulant. However, we are unable to redo/re-run the experiments as it will affect/deviate the original data. The authors take note on this comment for future study. 

Comparison of bio-coagulant dosages needed for kaolin suspension is suggested, for example comparing the efficiency to tannin coagulant.

Our response: The comparison of bio-coagulant dosages has been added in lines 416-423.

What is the charge density of the bio-coagulant?

Our response: The authors have added the value of charge density for the discussed bio-coagulant in lines 230-234.

Fig. 2 has only one graph, revise the caption.

Our response: The revision has been made.

Does this bio-coagulant leach some compounds to the treated water? Have you analysed TOC of treated water?

Our response: To answer this question, further experiment needs to be carried out. We take note on the possibility of the bio-coagulant in leaching certain organic and inorganic compounds. For this particular study, our scope is only on the turbidity removal. No TOC analysis was conducted. For the future study, detail analyses on chemical compounds in the treated water will be considered.

To discuss more deeply about mechanisms and possible restabilization, zeta potential or total surface charge analysis would be useful, if these are available for authors.

Our response: The authors thank the reviewer for the positive comments and suggestion. The authors have discussed on the relevant mechanism with regards to the protein content, as potential active coagulant agent, polyelectrolyte and surface charge in lines 226-234

Reviewer 2 Report

Dear Authors,

Thank you very much for giving me the chance to read and review your paper about the study of the bio-coagulant derived from papaya seed.

I have a couple of comments I hope can be helpful to improve the paper.

Comment 1

In paragraph 2.1  - Please, add information about the pH of the obtained solution

Comment 2

In chapter 2.2  -

According to the description of the bio-coagulant preparation the seed contained oil. The amount of oil in some papaya varieties is very high  up to 30%. My question is:  did the authors also examine defatted papaya seed as a raw material?

Is after preparation of the bio-coagulant solution the solid particle were removed?. What method was used. Please, rewrite the procedure for handling the obtained bio-coagulant solution.

Comment 3

In chapter 2.6  - please, add the information about the used software.

Comment 4 –

Please, elaborate information on the applied RSM method and the results achieved using this method:

 - in table 4 - add the results of all 33 runs,

 - in table 4 - the real levels -1 and +1  of independent variable “pH” are different from levels showed in table 2

 - Line 313 – 320 – please add information about fixed level of dosage. Please, explain why the turbidity removal [%]  achieves values above 100% on Fig. 4C

- What is the influence of the independent variables initial turbidity and pH on “coagulation activity”?. Is possible to add the figure showing this effect?

Comment 5–

Paragraph 3.1.3 - The surface morphology of the deshelled seed is important for the adsorption process. In relation to the scope of this article is only additional information.

Comment 6 - Technical errors:

  • figure 5 - line 374 - change the figure description; there is “turbidity removal” in my opinion should be “coagulation activity”.

Finally, the paper is ready for publication after revision

Author Response

Reviewer #2

Thank you very much for giving me the chance to read and review your paper about the study of the bio-coagulant derived from papaya seed. I have a couple of comments I hope can be helpful to improve the paper.

In paragraph 2.1: Please, add information about the pH of the obtained solution

Our response: The pH of the obtained solution has been added in line 94.

In chapter 2.2: According to the description of the bio-coagulant preparation the seed contained oil. The amount of oil in some papaya varieties is very high up to 30%. My question is: did the authors also examine defatted papaya seed as a raw material?

Our response: Many thanks for the question. Yes, we agreed that some papaya varieties contained high amount of oil. In our study, we did not examine the oil content of our deshelled papaya seeds. However, the deshelled process had removed certain percentage of oil.

Is after preparation of the bio-coagulant solution the solid particle were removed? What method was used. Please, rewrite the procedure for handling the obtained bio-coagulant solution.

Our response: This part has been revised in lines 104 -107.

In chapter 2.6: Please, add the information about the used software.

Our response: The name of the software has been added in lines 158-159.

Please, elaborate information on the applied RSM method and the results achieved using this method: Table 4 - Add the results of all 33 runs

         Our response: Table 4 represent the factorial analysis in obtaining the Pareto effect. Hence, it is only having 16 results (correspond to the 23 with duplication).

Table 4 - the real levels -1 and +1  of independent variable “pH” are different from levels showed in Table 2

Our response: Many thanks for the comment. The low and high range pH of 3.0 and 7.0 were used for initial factorial analysis to obtain the Pareto chart. For subsequent CCD analysis, minor revision was done on pH. The pH was slightly change to pH 4.0 and 6.5, taking into consideration of more feasible value for axial point (-a). If the low range of pH 3.0 was continuously used in the subsequent CCD analysis, the -a can be about pH 2.0, which then can potentially deteriorate the bio-coagulant. These explanations have been included in lines 167-171 and Table 2 has been revised as well.

Line 313 – 320 – please add information about fixed level of dosage. Please, explain why the turbidity removal [%]  achieves values above 100% on Fig. 4C

Our response: The information on the fixed level of dosage has been added. Regarding the Figure 4(c), the turbidity removal is not above 100%. The highest turbidity removal at that point was 93.1%. The y-axis was set at default thus, it set to 110%. 

What is the influence of the independent variables initial turbidity and pH on “coagulation activity”? Is possible to add the figure showing this effect?

Our response: Explanation on influence of the independent variables (initial turbidity and pH) on coagulation activity has been briefly added in lines 387-392. Figure 5(c) is also included to illustrate the effect.

Paragraph 3.1.3 - The surface morphology of the deshelled seed is important for the adsorption process. In relation to the scope of this article is only additional information.

Our response: The authors agree and thank you for the comment. The discussion is useful to clarify the surface morphology since this property is critical for adsorption process.

Figure 5 - line 374 - change the figure description; there is “turbidity removal” in my opinion should be “coagulation activity”.

Our response: The description has been changed. Yes, it is supposed to be coagulation activity.

Reviewer 3 Report

In this manuscript the authors used Carica papaya seeds as bio-coagulants and to treat turbidity water. Thanks to their biodegradable and environmental-friendly nature, papaya seed can replace chemical coagulants. The optimization of turbidity removal process and coagulation activity was achieved employing response surface methodology. The study is clear and well presented, and it can be accepted for publication in this form.

Author Response

Reviewer #3

In this manuscript the authors used Carica papaya seeds as bio-coagulants and to treat turbidity water. Thanks to their biodegradable and environmental-friendly nature, papaya seed can replace chemical coagulants. The optimization of turbidity removal process and coagulation activity was achieved employing response surface methodology. The study is clear and well presented, and it can be accepted for publication in this form.

Our response: The authors thank the reviewer for the very positive comment.

Round 2

Reviewer 1 Report

Authors have sufficiently improved the manuscript, and in my opinion, can be accepted for the publication.